# The Potential of Understory Production Systems to Improve Laying Hen Welfare

**DOI:** 10.3390/ani12172305

**Published:** 2022-09-05

**Authors:** Shaocong Yan, Chenyujing Yang, Lei Zhu, Yongji Xue

**Affiliations:** School of Economics and Management, Beijing Forestry University, Beijing 100083, China

**Keywords:** farm animal welfare, laying hens, cage system, free-range system, understory raising

## Abstract

**Simple Summary:**

Non-cage farming is gradually becoming the mainstream mode of poultry farming worldwide, which has led to concerns regarding the welfare of laying hens in China. Under huge pressure for the supply of eggs, China, with relatively insufficient land resources, is highly dependent on cage systems, thus posing significant challenges related to animal welfare. In the context of this dilemma, China’s abundant woodland resources provide a means to improve the welfare of laying hens, in particular, providing a wide living space for laying hens to express their natural behaviours, such as foraging and reproduction. At the same time, this profitable farming model has been welcomed and supported by farmers in some areas of China, and is gradually being promoted, which may provide a template and confidence for China and other countries to address the challenges of keeping hens in non-cage systems in order to improve animal welfare.

**Abstract:**

The welfare of laying hens in cage systems is of increasing concern. Represented by the European Union’s ‘End the Cage Age’ initiative, more and more countries have advocated cage-free farming. China, an important country for poultry farming and consumption in the world, is highly dependent on cage systems and lacks confidence in alternative (e.g., free-range) systems. In this context, using China’s abundant woodland resources (including natural forests, plantations, and commercial forests) to facilitate the management of laying hens in a free-range environment may provide highly promising welfare improvement programs. On the basis of the Five Freedoms, we assess the welfare status of understory laying hen management systems with reference to the behavioural needs and preferences of laying hens and the EU standards for free-range and organic production (highest animal welfare standards in the world). The results show that the considered systems meet or even exceed these standards, in terms of key indicators such as outdoor and indoor stocking density, outdoor activity time, and food and drug use. Specifically, the systems provide sufficient organic food for laying hens without using antibiotics. They allow laying hens to avoid beak trimming, as well as to express nesting, foraging, perching, reproductive, dustbathing and other priority behaviours. The presence of roosters and higher use of woodland space allow the laying hens to achieve better feather and bone conditions, thus reducing stress and fear damage. Notably, the predation problem is not yet considered significant. Second, there is evidence that understory laying hen systems are profitable and have been welcomed and supported by farmers and governments in the southwest, south, and north of China. However, whether it can be scaled up is uncertain, and further research is needed. In addition, laying hens in this management system face various risks, such as foot injury, parasitism, and high dependence on consumer markets, which must be considered. Overall, agro-forestry, or accurately, understory poultry raising, provides opportunities and possibilities for free-range laying hens and welfare improvement in China and other countries.

## 1. Introduction

Animal welfare and food safety have increasingly become a core pursuit in global agricultural development [1,2]. In areas such as Europe and America, animal welfare has become one of the main principles of agricultural development and is regarded as the key to ensuring the quality of animal products [3,4,5]. In 2021, the European Parliament expressed support for the ‘End the Cage Age’ initiative and strived to prohibit the use of cages for farm animals, including laying hens, by 2027. The European Commission responded by promising to make a legislative proposal by the end of 2023 to bring an end to the ‘caged chicken’ era [6]. 

Differences in welfare regulations between countries have led to the transfer of poultry production—particularly intensive farming—to countries with lower welfare standards [7]. At the same time, the non-cage culture of eggs in Europe and Australia has affected the egg import and sales system [8]. Many important international food companies and retailers use animal welfare as an important standard for marketing [9]. Therefore, China, which has much to improve on in terms of animal welfare standards, is expected to face many new challenges [10,11]. Taking advantage of lower production costs, poultry producers who mostly use the cage system in Europe and America may be tempted to transfer part of their production to China in the near future, which may further deepen the dependence of China’s poultry industry on cage systems [12,13]. However, China’s export of caged eggs may face increasingly severe animal welfare barriers [14]. Therefore, for China, farm animal welfare improvement through the implementation of ‘free-range laying hens’ systems deserves more attention than ever.

There is no doubt that the large-scale promotion of the ‘free-range laying hens’ management system and improvement of the welfare of farm animals will inevitably affect production and consumption in the livestock and poultry industries [15,16,17]. Therefore, the promotion of this system may be highly problematic for a country like China. Although non-cage farming is beneficial, in terms of improving the quality of poultry products [18], there are many uncertainties regarding its impact on egg production and supply at the national level. This is in conflict with the Chinese government’s goal of ensuring a stable supply of poultry products and basic self-sufficiency in production [19]. At the same time, all measures considered to improve animal welfare may also increase production costs [20,21,22]; for example, producing a dozen eggs from cage-free layers costs about 24 cents more than that from layers housed in conventional cages, comprising an increase in cost of production of about 36%, according to research conducted by the Coalition for Sustainable Egg Supply [9]. This is not appealing to most ordinary people, and bearing this burden in the long run may pose a problem. Controversy also stems from the fact that China must meet huge supply targets with limited land resources [23]. China is currently the world’s largest producer and consumer of poultry eggs. In the context of such large-scale production and consumption markets, it seems difficult to achieve the same output amounts (although the eggs may be of higher quality [24]) in the ‘free-range system’ model, and thus, the idea may be unacceptable in China. Furthermore, the ‘free-range system’ may put more pressure on land resources. China uses 9% of the world’s arable land, feeds 20% of the world’s population, and its per capita agricultural land resources are relatively insufficient [25,26]. Compared with the traditional cage system, the ‘free-range’ scheme requires many resources, especially land resources [27]; however, China’s agricultural modernization strategy takes intensification and scale as the development direction of animal husbandry [28], where the key goal is to use limited land resources to address problems associated with the production and development of animal husbandry. This, to some extent, means that the ‘free-range’ scheme conflicts with the current development strategy and land status in China.

Agro-forestry, the integration of poultry into woodlands [29], offers more opportunities and possibilities for the promotion of free-range farming systems in China. This is an interesting option for improving the welfare of laying hens. Much literature in agroforestry in recent years [29,30,31] shows that combining husbandry and forestry is an interesting development. This combination can increase the productivity of the land and diversify income sources for farmers. Several studies [32,33,34] suggest that the use of woodland for free-range poultry may improve their welfare. The current practice of and research on understory laying hen projects are not uncommon in China [35]; however, there has been little focus on animal welfare, and it is rarely seen as an alternative to cage farming systems. Therefore, the production potential of this farming model may be under-estimated at present. The ‘free-range laying hens’ management system has broad support and practical feasibility in China [36,37]. The high-quality egg products provided by this system not only meet the material pursuits of consumers [38], but also align better with their inner emotional identity [39] and conform to the development goal of the Chinese government to strengthen the supply capacity of green agricultural products [40]. In addition, understory laying hen farming may provide trees benefits such as nutrient cycling, weed management, and pest control [34,41].

In this review, we argue that forest land resources should be rationally exploited to gradually promote the free-range system of laying hens and improve their welfare in China. Agroforestry, or accurately, the integration of laying hens into woodland, offers opportunities and possibilities for welfare improvement. Whether it can be promoted on a large scale is unclear. Overall, it is crucial to improve the welfare of laying hens. 

## 2. Farm Animal Welfare

The Five Freedoms, as key assessment criteria for animal welfare [42,43], define five specific welfare improvement goals and highlight the corresponding five provisions [44]. In terms of farm animals, the domains of ‘freedom’ include nutrition (freedom from thirst, hunger and malnutrition), environment (freedom from discomfort and exposure), health (freedom from pain, injury and disease), behaviour (freedom to express normal behaviour), and mental state (freedom from fear and distress) [45]. With reference to the Five Freedoms, many scholars have quantified the welfare of farm animals in terms of their activity space, food, health, exercise, beak trimming, and natural expression [46,47,48]. This also provides reference for quantifying the welfare of laying hens in an understory laying hen project.

In recent years, consumers and the public have paid increasing attention to the welfare of farm animals [38]. People are increasingly interested in where their food comes from and how it is produced [49]. Studies have shown that when the animal welfare of land-based farm animals is compromised, there may be serious negative human health consequences due to the misuse of antibiotics, environmental degradation, and the consequences of intensification [50]. Thus, providing farm animal welfare information can stimulate appreciation and demand for related products [51]. For example, consumers may prefer to buy free-range eggs, as the hens are supposedly ‘happier’ and ‘healthier’, and they believe that such eggs taste better [52].

However, in China, animal welfare seems to be a relatively unfamiliar term. Under the pressure of relatively limited resources and large-scale supply and demand, intensification and scale are the key development direction of China’s poultry farming industry at present, as well as for the foreseeable future [53]. This means that the cage system will likely remain the focus of development. This seems to cast a shadow over the improvement of farm animal welfare in China. Moreover, there is no legislation in mainland China regarding farm animal welfare, particularly in relation to laying hens, and only some elements of animal welfare (e.g., disease prevention and control) are included in production safety and animal health legislation [54,55]. The World Animal Protection Association has rated China’s Animal Protection Index (API) legislation on farm animal protection as G, the lowest level [11]. Therefore, on a practical level, the issue of farm animal welfare represented by ‘free-range laying hens’ has not yet become a focus of China’s agricultural development strategy [53]. At the theoretical level, farm animal welfare research in China is still in its infancy [37]. However, it has been found that there are benefits associated with improving the welfare of laying hens. For example, giving laying hens greater behavioural opportunities and freedom of movement can reduce their stress and fear, destructive behaviour and, therefore, mortality [56]. At the same time, environmental enrichment can reduce severe feather pecking and strengthen the immune system of laying hens, thereby reducing losses [57]. In addition, consumers often associate improved laying hen welfare with food health and are willing to pay higher prices for such products [58]. Overall, economization and marketization are generally positive for farm animal welfare, as manufacturers and retailers compete for the added value created by welfare standards and labels [59].

## 3. Challenges and Solutions to the Free-Range System

In general, a vast number of developing countries, including China, face many challenges in the large-scale implementation of the ‘free-range laying hens’ system [60]. This is not only due to resources—especially land resources—but also to a lack of confidence. First, land resources pose a challenge. Although China’s agricultural land (i.e., arable land, garden land, woodland, and grassland) resources are relatively sufficient, totalling about 7 million square kilometres [61], the per capita share is small: less than one-third of the world’s average. Moreover, with the continuous advancement of urbanization in China [62], poultry farming land resources continue to decrease. Most importantly, laying hen farming in China faces the hard constraints of the ‘red line’ of 1.8 billion acres of arable land and ecological environmental protection [63], as well as land competition due to pig farming [64], which further aggravates the land dilemma of promoting the ‘free-range laying hens’ farming model.

Second, free-range systems are associated with higher cost than caged systems. The European and American laying hen free-range system mainly reflects the characteristics of a longer raising cycle, a smaller hen-house density, a larger activity space, and more organic and green feed [65,66,67]. This necessitates greater time, space, and material costs, which are finally transmitted to the consumer side and reflected in higher prices [68,69]. Whether consumers can bear this cost in the long term still needs careful investigation. This may be due to the small production scale and low yield of the ‘free-range laying hens’ system. In China, the cage farming system and intensive farming are the mainstream systems used in the livestock and poultry industries (Figure 1a). The ‘free-range’ model—or welfare farming—is small in size and seems only subordinate to strengthening the supply of green agricultural products [70]. In 2020, China’s green food production of eggs and poultry meat was only 214,000 tons, accounting for less than one per cent of the total production, with opportunity for expansion [71].

In addition, the attention given to this topic to date has not been sufficient. Based on the above analysis, China’s laying hen free-range system lacks legal protection, and there is also a lack of targeted policy planning. Meanwhile, only a few producers have voluntarily implemented such systems [72]. According to field research, the free-range system of laying hens of local farmers is mainly used to meet their own needs and subsidize household use, being mostly scattered and non-systematic. This indicates that China lacks confidence in the free-range system, particularly under the huge supply pressure.

Thus, solving the problem of land resources and balancing supply and demand is the key to improving China’s implementation of the free-range system for improving laying hen welfare. Several studies have shown that agro-forestry, the incorporation of animals into woodlands, provides reciprocal benefits for both trees and animals, such as nutrient cycling, natural conservation, weed and pest control [30,33,34]. This is in line with the expectations and requirements of the Chinese government for woodland use [73]. It may provide benefits in terms of animal welfare while supporting ecosystem services and reducing environmental degradation. According to the statistics, China’s under-forest economic land area exceeded 400,000 square kilometres in 2021 [74], widely distributed throughout the country, providing a possible opportunity that can be taken advantage of to improve the welfare of laying hens (Figure 1b). The ownership of forest land resources in China belongs to the state and the collective, while the contracting right belongs to farmers [75]. Therefore, the premise of carrying out an understory laying hen project is to obtain the permission of the government and the village collective. This permission stipulates that the use of forest land must conform to the principles of protecting the ecological environment and sustainable utilization of forest land [73]. At the same time, livestock and poultry farming is prohibited in the core areas and buffer zones of nature reserves [76]. This means that an understory laying hen project cannot have a negative impact on the forest environment, especially in terms of avoiding the protection area of endangered species. Such restrictions and regulations objectively improve the welfare of laying hens. For example, regarding the need of forest environmental protection, there is an upper limit (2500–3000/ha [77,78,79,80]) to the number of laying hens per hectare. This objectively expands the space for each hen to move freely and alleviates the frustration caused by a lack of space [81], leading to a significant improvement in their welfare. At the same time, the forest land may contain abundant natural food, which is beneficial for laying hens to show normal foraging behaviours and avoid being subjected to beak trimming, which is common in cage systems [82]. Most importantly, considering the most critical woodland resources for understory laying hen system, the majority of farmers have relatively low use costs. In addition, China’s diverse forest types provide more options for understory laying hen system and more possibilities for improving the welfare of laying hens.

## 4. Dawn of the Implementation of the Free-Range System: Understory Raising

As a model free-range system, understory laying hen farming is in line with consumer perceptions of animal welfare. According to the general perception of consumers, free-range and organic-cultured eggs are healthier and better in quality than those produced in caged systems [83], and thus, consumers are willing to pay a higher premium for them [84].

Egg production under the cage system does not seem to show outstanding advantages [85]. Studies have shown that the higher the cage density in an intensive farming system, the greater the decline in egg production and quality [86,87]. Moreover, it has been found that a longer duration of high cage density leads to a lower egg production rate in laying hens [88]. The daily egg production of free-range hens (89.27%) is significantly higher than that of conventional cage (87.1%) and enriched cage (87.26%) hens. At the same time, the egg mass of hens in a free-range system was higher by 5.21% and 5.47%, compared with conventional cage and enrichment cage hens, respectively [89]. This is in conflict with the traditional impression of the free-range system of laying hens; that is, producing eggs in low yields but with high quality [90,91,92].

Free-range eggs have a certain consumer market. Internationally, the demand for organic and non-cage eggs in developed countries such as Europe and the United States has grown rapidly. The number of individuals in the United States consuming organic eggs increased by 14% over the span of 2014–2017, reaching 80.35 million [93]. In 2019, 53% of eggs came from free-range systems and nearly 60% from non-cage systems in the United Kingdom [94]. More than 2000 companies (covering catering, hotel, fast-moving consumer goods, and other fields) have promised to buy 100% non-cage eggs worldwide, and the commitments of more than 50 companies include China. This means that understory laying hen projects could serve as the beginning of a ‘blue ocean’ market [95]. Some Chinese consumers prefer eggs with the ‘organic’ and ‘free-range’ labels and are willing to pay a price premium for them [96].

Specifically, understory laying hen farming projects mainly rely on woodland resources belonging to the government and the collective. This means that investment in the protection and management of forest land resources comes mainly from local village groups, relieving producers of the initial costs associated with implementing these projects. Furthermore, the producers only need to pay low costs, in terms of forest land use rights [97]. At the same time, woodland is the main outdoor space and an important food source for laying hens. This greatly reduces the investments in feed and fixed assets in the project [98]. These facts show that the understory laying hen system has a certain profit space and attractiveness.

In addition, the understory laying hen project provides more opportunities to improve the welfare of laying hens. The access of laying hens to woodlands increases the likelihood that they will express a wider range of normal behavioural patterns [99]. The behavioural needs of laying hens (nesting, foraging, increased space, perching, dustbathing and other behaviours) have different preferences and priorities [100]. The degree to which these needs are met reflects the welfare of the laying hen. Food is valuable to hens and is an important criterion [101]. Pre-laying (nesting) behaviour is given high priority. As oviposition approaches, laying hens have a strong preference for a discrete, enclosed nest site and access to a nest site is even a priority over food. Foraging is a priority and necessary behaviour. Semi-wild junglefowl hens (similar to the behaviour of laying hens) spend 60% of their active time of the day on foraging behaviour [102]. This can be used to assess the welfare of laying hens. Perching and dustbathing are behavioural needs, and it is unclear how much they are valued by laying hens. There is evidence [103,104] that laying hens prefer personal space and exhibit spacing behaviour. Hens need additional space for specific activities such as flapping their wings and reducing negative social interactions [105]. At the same time, reduced stocking density reduces feather pecking and aggression [106]. In addition, considering that the access to outdoor areas for laying hens may increase the risk of predation [107], farmers usually set up fences and henhouses to prevent the invasion of predators [108,109]. On the basis of the Five Freedoms [45], we assess the welfare status of understory laying hen management systems with reference to the behavioural needs and preferences of laying hens and the standards of free-range systems and organic farming in the European Union (highest animal welfare standards in the world [48]), the United Kingdom, Australia and other countries or regions [110,111,112]: first, food and water, and whether the feed is adequate and organic; second, the perch of laying hens, which includes the form of the henhouse, furniture, and indoor stocking density; third, whether drugs and antibiotics are used [113]; fourth, whether stress and fear damage can be avoided or reduced; studies have shown that more space and freedom of movement can reduce stress and fear in laying hens [56]; in addition, the presence of roosters can reduce the fear of laying hens [114]; fifth, whether the laying hens can express normal behaviour; this is mainly reflected in the degree to which the behavioural needs of the laying hens are met. Therefore, whether the beak is trimmed [115], outdoor stocking density and activity time need to be taken into consideration. As far as stocking density is concerned, the maximum indoor stocking density is set at nine chickens per square meter by the EU Directive 1999/74 (minimum standards for the welfare of laying hens), and the outdoor activity space should consist at least in 4 square meters per chicken, with continuous outdoor roaming allowed during the day. [116,117,118].

We studied three cases of free-range laying hens using different woodland types (bamboo forest, plantation forest and wolfberry forest) to more fully analyse the extent to which this model improves the welfare of laying hens.

### 4.1. Raising Laying Hens in a Bambusoideae (Bamboo) Forest in Lujiang County

China has the most abundant bamboo forest resources in the world, which are widely distributed and mainly concentrated south of the Yangtze River. The area of bamboo forests in China reaches 6.4 million hectares [119], which provides the basic conditions for the development and promotion of the bamboo forest laying hen system.

Fanshan Town of Lujiang County is located in Hefei City, Anhui Province. Its bamboo resources cover up to 1333 hectares, with a forest coverage rate higher than 75%. Under the guidance of large chicken farmers, Fanshan Town has been developing a moderate-scale bamboo forest laying hen system (Figure 2a), with the aim of making Fanshan free-range laying hens a national product of geographical importance [120]. Raising laying hens in bamboo forests is a near-natural operation that keeps laying hens near an ideal state of animal welfare [121]. First, the living space is spacious, as in terms of outdoor activity space, the stocking density of laying hens in bamboo forests is generally 2200–2500/ha. Each hen has an activity space of 4–4.5 square meters and has more than 9 h of outdoor activity during the day. This respects the hen’s preference for private space and provides much of the space and time needed to express foraging behaviour [100,102]. Adequate exercise improves the bone and muscle condition of the laying hens [122,123]. Laying hens normally express dustbathing behaviour in bamboo forests to increase pleasure [124]. At the same time, the producers have built sheds for the laying hens in the bamboo forest (Figure 2b). A 500 square-meter sheds can generally accommodate 3500–4000 laying hens, resulting in an indoor stocking density of 7–8 laying hens per square meter. These sheds have a trapezoidal long wooden bed to facilitate the free indoor movement of the laying hens. The shed is equipped with discrete and multi-level nests to allow for access to more hens. This also facilitates the producer to collect eggs manually. Second, the laying hens are fed more humanely, as most of the food requirements of the laying hens are satisfied by foraging freely in the bamboo forest, and they are supplemented with organic feed such as grains and corn (accounting for about 30% of their total food intake). As the hens need to hunt insects and feed on weeds freely in the bamboo forest, beak trimming is avoided. Free foraging reduces feather pecking in laying hens [125]. Third, no drugs (especially antibiotics) are used in the laying hens rearing stage, although completely banning drug use is not entirely positive for the welfare of laying hens. Fourth, in this model, roosters and laying hens are reared together, and the laying hens exhibit normal reproductive behaviour. At the same time, the protective role of the rooster reduces the risk of the laying hens being frightened in the bamboo woodland and increases the activity of the flock.

The project mainly depends on the local natural bamboo forest resources, requiring less investment in rearing facilities and fixed assets. Natural food resources in woodlands reduce feeding costs. The presence of laying hens also provides benefits such as nutrient cycling for bamboo forests [126]. A laying hen under this management system is a dual-purpose breed with meat value. It starts laying at 150 days old and lasts 5–7 months before it is sold for meat. Each hen produces an average of 140–160 eggs. The annual net income brought by each hen is more than CNY 60 and, in the local project, the annual output value of the native laying hens alone has reached CNY 150 million. Furthermore, a ‘chicken–bamboo industry–tourism’ industrial chain has been formed, receiving 2000–3000 tourists per day. Natural bamboo forest resources are widely distributed in southern China, where an important consumer market for free-range eggs is also located [127,128]. Therefore, a bamboo forest laying hen system is suitable for promotion in the southern region of China. According to incomplete statistics, the project has been widely welcomed by local farmers in Guangdong, Guangxi, and Zhejiang [129].

### 4.2. The ‘Forest–Grass–Chicken’ Ecological Farming Scheme in Fangshan District

China’s policy of returning farmland to forests has opened up more forest land resources for free-range laying hens, with a total of 34.8 million hectares of farmland having been returned to forests over the past 20 years [130]. In addition, due to the need for ecological protection, the effective development of these forest and grass resources is low. On this basis, the ‘forest–grass–chicken’ ecological farming model in Fangshan District of Beijing can be feasibly promoted, as the government has assumed most of the forest land maintenance costs.

Beijing You Chicken (BYC) is a meat/egg dual-purpose native chicken in Beijing. It is famous for its unique appearance and high-quality eggs and has been listed as one of the most important chicken breeds by the Ministry of Agriculture and Rural Affairs of China [131]. Many companies and farms use orchards, slopes, and forests to give BYCs more space, leading to significant economic benefits [132].

The terrain of Fangshan District in Beijing is relatively flat, and its forest coverage rate reaches 36.9%. In recent years, a total of 10,867 hectares of artificial afforestation, 9333 hectares of closed hillsides, and 30,000 hectares of tended trees have been established [133]. This has effectively expanded the land resources for understory laying hen projects. The ‘forest–grass–chicken’ ecological farming scheme has significantly improved the welfare of BYCs. First, the stocking density of laying hens in woodland is lower, compared with the bamboo forest laying hen system. In this scheme, hens are stocked in small groups of low density (1500–1800/ha) in woodland. The outdoor activity space of each hen is no less than 5.56 square meters, so that they have enough space to express spacing behaviour [104]. With more than 9 h of continuous outdoor activity during the day, laying hens have a sufficient time budget for foraging activities, thus reducing the incidence of feather damage [125]. At the same time, small-scale scattered mobile houses have been scientifically established, according to tree species, spacing, canopy density, and other factors (Figure 3a). Each mobile house is up to 13 square meters and can accommodate 65–100 laying hens. The indoor stocking density is 5–8 laying hens per square meter. The mobile house is equipped with a full set of facilities, allowing for activities such as eating, drinking, habitation, and egg-laying. Four nests are placed in a mobile house and covered with soft materials such as rice husks to facilitate more laying hens for pre-laying behaviour and egg production. A special place for dustbathing is set up next to the mobile house. Second, their food is mainly composed of grass, vegetables, and insects, supplemented by organic feed. This ecological farming scheme seeks to achieve self-sufficiency, in terms of food for the hens. High-quality artificial grassland, characterised by high production performance and good palatability for chickens, as well as pure natural green vegetables, are planted in the forest land as the main food sources for laying hens. It is a free dietary choice. This requires keeping their beaks intact, allowing them to forage freely in the woodland (Figure 3b). Third, the scheme minimizes the use of drugs and antibiotics. Finally, the ‘forest–grass–chicken’ ecological farming scheme places roosters and laying hens together. The presence of roosters significantly reduces the duration and incidence of tonic immobility in laying hens, and reduces the damage caused by stress and fear [114].

Due to the proximity of the Beijing–Tianjin–Hebei consumer market, the economic benefits of this farming model can be maximized. Under this management system, the raising cycle of laying hens is 500 days. Producers need to train hens to lay eggs in fixed nests and manually collect eggs daily. A hen starts laying at 150 days old and lasts for 10–12 months. Each hen produces an average of 120 eggs and is sold for meat at the end of production. According to statistics, the annual net income of each hen is more than CNY 50–100, achieving a local forest income of CNY 75,000–150,000 per hectare. The outstanding economic benefits are also promoting this model in the areas where farmland is being returned to forests, such as Hebei Province [134].

### 4.3. Raising Laying Hens in a Lycium chinense Miller (Wolfberry) Forest in Wuzhong City

Northwest China’s arid and semi-arid area accounts for 30% of the national land, but only 4% of its population, which means that the local land resources are relatively abundant. Hongsipu District, Wuzhong City, Ningxia, is located in northwest China. The climate is arid, and water resources are relatively abundant, but the ecological environment is fragile, which makes it unsuitable for large-scale intensive farming. This area also has important wolfberry plantations, with the planting area reaching 3733 hectares. Therefore, the development of a poultry production system based on wolfberry plantations has become a feasible possibility (Figure 4a). This provides a realistic scenario for promoting the free-range system and improving farm animal welfare in the arid areas of Northwest China.

First, this farming model provides a wide outdoor space for the native laying hens. The density of native laying hens in wolfberry woodland is 1200–1500 per hectare, and their outdoor activity time is about 8 h. Laying hens have plenty of time for foraging activities in the wolfberry woodland during the day. The activity space of each hen is 6.6–8.3 square meters, well above the set welfare standards and comprising the lowest density of the three systems discussed here. This provides enough private space for laying hens to express comfortable behaviour [104,135]. Extensive outdoor space relieves stress and fear of laying hens [136]. At the same time, the wolfberry forest provides a place for dustbathing and shade for the native laying hens, which protects them from the negative effects of the high temperature and high intensities of ultraviolet radiation common in northwest China [137,138]. Second, the indoor stocking density in this mode is also relatively low. The henhouse is in the form of a polytunnel shed (Figure 4b). A 500 square-meter polytunnel shed generally accommodates 3000 laying hens, with six hens per square meter. Perches and nests are placed in the polytunnel shed to facilitate free movement and pre-laying behaviour of laying hens. In addition, the food of the native laying hens is mainly composed of natural food in the wolfberry plantations, supplemented with organic feed (in a ratio of about 6:4). Therefore, native laying hens mainly feed on wolfberry fruit, wolfberry leaves, weeds, insects, and other natural food. This provides the benefits of weed management and pest control for wolfberry forests. Their beaks are not trimmed, allowing the laying hens to exhibit normal foraging behaviours. Foraging activities increase the use of woodland space by laying hens, both reducing the risk of severe feather pecking [139,140]. Finally, similar to the treatment of Fanshan native laying hens, the use of any drugs during the rearing of the native laying hens is prohibited. In this model, roosters and laying hens are also mixed. The hens show normal behaviour such as reproduction and communication, as well as fewer signs of fear. The laying hen under this management system has a meat value. It starts laying at 150–180 days old and lasts for 8–10 months, after which it is sold for meat. Each hen produces an average of 130–150 eggs. The unit price per egg is more than twice that of the average local egg. Locally, the unit price of these eggs is more than twice that of ordinary eggs. This fact also increases the attractiveness of implementing free-range schemes in other types of commercial forests and plantations. Therefore, even in China’s arid northwestern region, understory space is still an ideal place to improve the welfare of laying hens.

## 5. Discussion and Conclusions

### 5.1. Understory Laying Hen Projects Objectively Improve the Welfare of Laying Hens

The understory laying hen system objectively provide hens with a relatively ideal and natural state. As shown in Table 1, all three management systems considered herein significantly improve the welfare of the laying hens. Firstly, the systems provide sufficient natural food and organic feed for laying hens to maintain full health and vigour. Secondly, the systems all prohibit the use of antibiotics. Thirdly, sheds, mobile houses and polytunnel sheds provide perches for laying hens. Furniture in the henhouse, such as nests, helps meet the behavioural needs of laying hens. Hens are able to express nesting, foraging, perching, reproductive, dustbathing and other priority behaviours [100,104].

Specifically, all three systems reached or exceeded the minimum standards of laying hen welfare required by the European Council [141]. In particular, the outdoor stocking densities for all three systems are higher than the minimum standard of 4 square meters per hen. The outdoor stocking density in the wolfberry plantation hen management system reached 1200–1500/ha, close to the free-range standard of EU laying hens (≤1000/ha) [142]. Laying hens are able to express spacing behaviour normally [104]. The indoor stocking densities under all three systems were below the maximum limit of nine hens per square meter (EU). More space and freedom of movement can reduce stress and fear in laying hens and reduce destructive behaviour, thus reducing mortality in laying hens [56]. The outdoor activity time of the laying hens in all three systems reached or exceeded 8 h, which is the welfare standard of free-range laying hens stipulated by the Royal Society for the Prevention of Cruelty to Animals (RSPCA) [143]. This provides a sufficient time budget for foraging activities. The laying hens in all three described systems mainly relied on foraging freely for insects, weeds, fruits, and so on in the forest. Therefore, their beaks were kept intact, ensuring their normal foraging behaviour. These increase the use of woodland space by laying hens, which in turn reduces the risk of severe feather pecking [144]. The woodland is also a place for dustbathing. The laying hens also have better feather and bone conditions. In addition, the systems provide free dietary choice for laying hens to promote their foraging activities, thus improving animal welfare [145]. Finally, putting roosters and laying hens together reduces the incidence of fear and feather damage in laying hens, and the presence of a rooster broadens the behavioural repertoire of laying hens [146]. Notably, predation was not yet a key concern. Fences and aviaries effectively prevent predators from entering. In addition, as the woodlands used are generally close to villages or towns, the hens tend to encounter fewer natural predators in the wild.

In general, understory laying hen management systems provide more opportunities and possibilities for improving the welfare of laying hens, and basically achieve the welfare goals of the Five Freedoms. In terms of major indicators, they match or even exceed the standards of the EU free-range system and meet the priority behavioural needs of laying hens, thus placing them at a high level of welfare.

We realize that the starting point of raising laying hens in forests is to increase income, rather than to improve animal welfare. Its essence is still the use of animals for profit. We recognize that it is difficult to convince farmers to actively promote the free-range system or improve the welfare of laying hens from the perspective of ethical considerations, as this typically entails higher costs and more risks [147]. In addition, many farmers are concerned about higher mortality and more serious health problems for laying hens in non-cage systems [148,149]. Compared with farmers, consumers and retailers have a more active role in promoting free-range systems and improving the welfare of laying hens [38,59]. Especially for consumers, understory raising projects meet their requirements, in terms of both animal welfare and food quality [52,150]. Meanwhile, more and more Chinese consumers are willing to pay for animal welfare-promoting products [151]. This drives retailers to follow animal welfare standards and use labels to create additional value [152]. Further, at the request of consumers and retailers, producers must improve the welfare of their laying hens to obtain better prices for their products and improve access to the market [153,154].

### 5.2. Rational Development of Forest-Land Resources Is an Effective Way to Solve the Dilemma of Cage-Free Laying Hen Management Systems

An appropriate scale of understory ‘cage-free laying hens’ is expected to not only help promote the virtuous cycle of local forest ecosystems, but also activates unutilized understory resources and enables increased incomes for farmers. The integration of laying hens into the woodland provides reciprocal benefits for both trees and hens. Laying hens are at a high level of welfare while the trees gain benefits such as nutrient cycling and pest control. Moderate scale is the key to achieving these goals. However, this project is still in the initial stage of exploration in China, and an appropriate stocking density has not yet been formed. This constitutes our future research focus. Overall, this management system, which takes ecological, economic, and social benefits into account, provides strong support for improving the welfare of laying hens and promoting the free stocking system.

The three cases selected in this review were bamboo forest laying hen management systemin the middle and lower reaches of the Yangtze River in China, raising laying hens based on forests reclaimed from farmland near the Beijing–Tianjin–Hebei consumer market, and raising laying hens in wolfberry plantations in arid northwest China. These three regions are fairly representative of the country, being geographically located in the north, south, and northwest of China, respectively, and also represent the different levels of economic development in different regions of China. This reflects the universality and diversity of the understory laying hen management system and the potential for its promotion. Although the three systems present their own particularities, they all achieve a balance between improving the welfare of laying hens and economic benefits. In particular, the latter is the key to popularizing the project. There is evidence that understory laying hen projects are welcomed and supported by farmers and governments in the southwest, south, and north of China [155,156,157]. For example, Lincang City in Yunnan has nut groves and coffee forests, Fengjie County in Chongqing has mulberry gardens, and Moyu County in Xinjiang has walnut groves, all of which have been used to build laying hen free-range systems [158,159,160].

Therefore, we believe that under the premise of ecological protection, the organic embedding of the ‘free-range laying hens’ farming model in woodlands is a feasible way to implement the free-range system and to improve the welfare of laying hens in China. In the future, we intend to further study the impact of large-scale implementation of this model on China’s egg supply system.

### 5.3. Improvement of Animal Welfare Driven by the Understory Laying Hen Scheme Needs a Dialectical View

At heart, an understory laying hen management system stems from the pursuit of economic benefits. Its essence is to bring higher economic benefits, along with higher laying hens welfare. In other words, it is an economic benefit-oriented improvement scheme for farm animal welfare. Its core is still an economic-driven one and, as such, does not fully embody the idea of animal welfare. One must take into account that the future of such models may also fluctuate with the economy, a pattern that it is not perfect.

First, this project is not suitable for areas far from the consumer market or economically under-developed areas. For example, the wolfberry plantation laying hens project in China’s northwest arid area mainly depends on the eastern consumer market. High transportation logistics costs and limited channels of market sales directly weaken its economic benefits, especially in periods of market volatility (e.g., epidemic shocks) [161]. This directly reduces the enthusiasm of farmers with respect to improving the living conditions of laying hens, which may lead to a worsening in the welfare of the laying hens.

In addition, there are certain risks and costs in such projects. Several studies have shown that an average of 3.7% of hens in a free-range flock are estimated to have died from predation [162]. Our research in China also found this problem. For example, in an interview with farmers in Horqin Right Front Banner, we found that foxes were the main predators of free-range laying hens; however, overall, predation did not seem to be a key concern. The fences and aviaries provide shelter for the laying hens, preventing predators from entering. At the same time, hens encounter fewer natural enemies in the wild, as they are near villages or towns. Relevant research has also shown that predation is not common in adult chickens [163]. Moreover, hens in the free-range system usually have higher rates of foot injuries and parasite infections, due to additional activity spaces [89,122]; however, this needs to be viewed dialectically. Compared with traditional caged hens, the bone and muscle strength of those in free-range systems is significantly higher, due to adequate access to movement space [140]. Therefore, the cost–income ratio of understory laying hens management systems should be further quantified in the future.

However, in general, such projects serve as a key step for developing countries, in terms of improving the welfare of laying hens. Farmers are regarded as the main players to improve farm animal welfare. Therefore, improving the welfare of farm animals needs to consider the associated impact on the interests of farmers [164]. Free-range hens can provide farmers with a livelihood, food security, income, and other social and cultural obligations [165]; however, a breakthrough has largely been achieved in China, as farmers have begun to consider whether their hens live in spacious areas, whether they eat well, whether they are well rested, and whether they can freely express their nature and reproduce [54]. Such issues were rarely considered even a few years ago.

## 6. Conclusions

Although it is not clear whether understory laying hen systems can be promoted on a large scale, it is clear that agro-forestry provides more opportunities and confidence for the implementation of free-range laying hens in China. The integration of laying hens into the woodland is interesting and may bring benefits in terms of improved welfare and protection of the woodland environment. We hope that these practices will provide inspiration and confidence for China and other countries to address the challenges of keeping hens in non-cage systems

## Figures and Tables

**Figure 1 animals-12-02305-f001:**
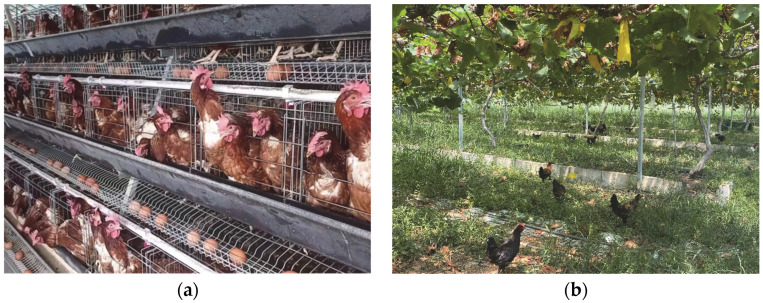
Production of laying hens and eggs in China. (**a**) Intensive cage system for laying hens. (**b**) Raising laying hens in the forest. Note: (**a**) provided by Shaocong Yan; (**b**) provided by Lei Zhu.

**Figure 2 animals-12-02305-f002:**
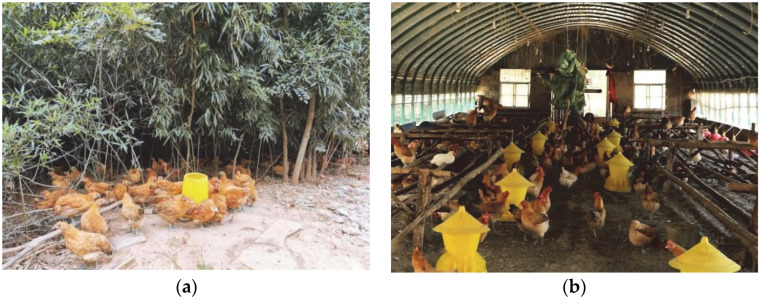
Raising laying hens in a bamboo forest. (**a**) Laying hens resting and drinking in a bamboo forest; (**b**) A shed with perches, feeders and drinkers. Note: (**a**) taken by Shaocong Yan; (**b**) from Lujiang News Network: http://www.ahljnews.com/8229381/33969499.html (accessed on 15 June 2022).

**Figure 3 animals-12-02305-f003:**
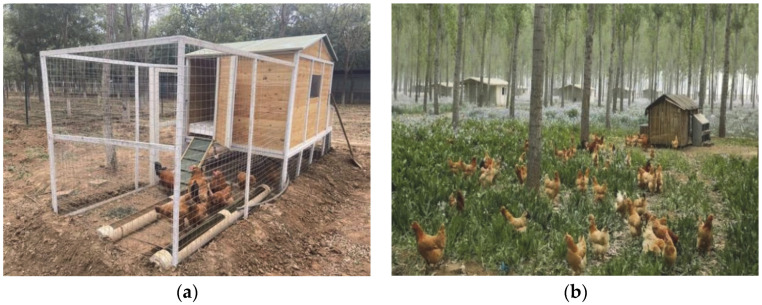
The ‘forest–grass–chicken’ ecological farming scheme. (**a**) The small and low-density mobile house. (**b**) Laying hens foraging freely in woodland and grassland. Note: (**a**) from the Official Network of the People’s Government of Fangshan District, Beijing: http://www.bjfsh.gov.cn/zhxw/fsdt/202010/t20201009_40007024.shtml (accessed on 4 June 2022); (**b**) from Beijing Agriculture and Rural Bureau: http://nyncj.beijing.gov.cn/nyj/snxx/gzdt/10937948/index.html (accessed on 8 June 2022).

**Figure 4 animals-12-02305-f004:**
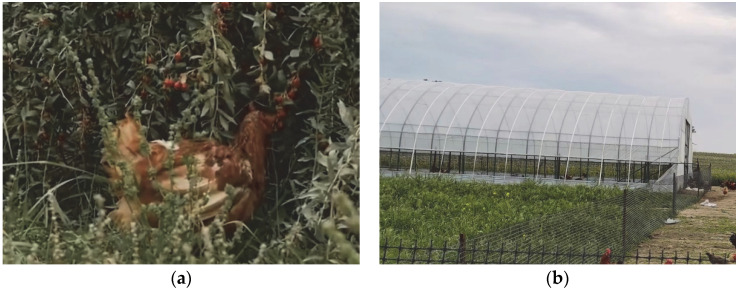
Raising laying hens in a wolfberry forest. (**a**) Native laying hens in the wolfberry plantation. (**b**) A polytunnel shed. Note: (**a**) from China Network Television (CNTV): https://tv.cctv.com/2014/09/11/VIDE1410371397626890.shtml (accessed on 6 August 2022); (**b**) taken by Shaocong Yan.

**Table 1 animals-12-02305-t001:** Characteristics of three understory laying hen management systems.

System	Raising Laying Hens in the Bamboo Forest	The ‘Forest–Grass–Chicken’ Ecological Farming Scheme	Raising Laying Hens in the Wolfberry Forest
**Forest type**	Bamboo forest	Artificial forest and artificial grassland	Wolfberry forest
**Outdoor stocking density**	2200–2500/ha; Each hen has an activity space of 4–4.5 square meters	1500–1800/ha; Each hen has an activity space more than 5.56 square meters	1200–1500/ha; Each hen has an activity space more than 6.6 square meters
**Outdoor activity duration**	≥9 h	≥9 h	About 8 h
**Housing**	Shed	Mobile house	Polytunnel shed
**Fence**	Yes	Yes	Yes
**Indoor stocking density**	7–8 hens per square meter	5–8 hens per square meter	About 6 hens per square meter
**Food composition**	Natural food accounts for 70% and organic feed accounts for 30%	Artificially planted grasses, vegetables and forest insects. Supplemented by organic feed	Natural food accounts for 60% and organic feed accounts for 40%
**Free dietary choice**	Yes	Yes	Yes
**Drugs use**	No	Try not to use	No
**Antibiotics**	No	No	No
**Perch**	Yes	Yes	Yes
**Nest**	Yes	Yes	Yes
**Beak trimming**	No	No	No
**Free foraging**	Yes	Yes	Yes
**Keeping roosters**	Yes	Yes	Yes
**Dustbathing**	Yes	Yes	Yes
**Spacing behaviour**	Yes	Yes	Yes
**Onset of lay**	150 days old	150 days old	150–180 days old
**Duration of the laying period**	5–7 months	10–12 months	8–10 months
**Egg production**	140–160 eggs/bird	120 eggs/bird	130–150 eggs/bird
**Meat value**	Yes	Yes	Yes
**Annual net income per hen**	>CNY 60	>CNY 50	>CNY 50

## Data Availability

Not applicable.

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
