# Peer review of "The Potential of Understory Production Systems to Improve Laying Hen Welfare"

_animals, 2022, doi:10.3390/ani12172305_

Round 1

Reviewer 1 Report

First, I really enjoyed reading the manuscript. The authors are concerned with the conditions of poultry production in China, proposing alternative methods based on three different forest-based systems, aiming above all the improvement of animal welfare. The three proposed systems, Bamboo Forest, 'forest–grass–chicken' and Wolfberry Forest, are clearly promising, despite some limitations that the authors mention.

My main concern with this review is mainly with the writing style, which is more geared towards a magazine like National Geographic than a scientific journal.

When I read the authors' affiliation, School of Economics and Management, I thought this review would focus more on the economic viability of these systems, but this approach doesn't go very deep. In the literature, there are several articles and books on the economic viability of free-range systems, with some data that could be included in this review.

As for the title, perhaps too ambitious, perhaps a bit vague. I would include expression such as "forest-based" and "free-range systems".

Between lines 49 and 53, specify "transfer of poultry production" and the "impacts” on “the domestic livestock and poultry market in China".

In lines 61 and 62, explain why "the promotion of this system in China is highly controversial".

Line 72, "overall export volume" of what?

Line 75 refers to "foster care model" but I couldn't find any reference to this model.

Lines 85, 102-103, 165, 193, 350 and 538, I think the term "contradictions" is not being used properly.

Lines 96-97, I read the referred article (#24 - Goldberg, 2016), full-text, and didn't find any reference about the Chinese government.

Lines 111 and 112, refer to export volumes per ton (c) and USD (d).

Lines 116-130, as this paragraph doesn't add much, I would delete it.

Lines 141 and 167, "aquaculture", poor translation or typing error?

Line 320, "better enjoy their environment", expression perhaps too subjective.

Line 323-325, rewrite to clarify the meaning of the text.

Line 337-339, rewrite to clarify the meaning of the text.

Line 358, "wolfberry chicken", use alternative designation as it appears to be referring to a breed of chicken.

Line 360, “breeding” instead of “cultivation”.

Line 360, I googled "Meiqi chicken" but got no results.

Line 418, the term "happier" is debatable.

Lines 452-454, from "More" to "welfare", this statement should be better substantiated.

Lines 502-504, from "As gatekeepers" to "welfare", rewrite to carify the meaning.

Reviewer 2 Report

Dear Authors,

Thank you for submitting this paper that investigates the use of forest habitats as a farming measure for chickens. The study is thought-provoking and has the potential to stimulate improvements in animal welfare.

At current however, there seem to be some large revisions required in the manuscript to ensure the work is scientifically robust. I have attached the PDF version of the manuscript with specific comments. Additionally, please consider the following points: 

1. Welfare. How can welfare be quantified? Is there any evidence that chickens in this management regime are actually better off?

2. How will chickens be kept in an area? Are aviaries or fences required?

3. Disease and predation? These are major concerns as they could result in animal welfare and loss of individuals? Is there any evidence to demonstrate these are small risks?

4. What else would have featured in these areas? Is there a risk to endangered species?

5. What is the appropriate stocking density? What happens if this is exceeded?

Round 2

Reviewer 1 Report

This review is very meritorious in the sense that it proposes production systems that aim to improve the welfare of laying hens in the world's largest producer and consumer of poultry meat and eggs.

Therefore, and to contribute to this evolution in the of animal welfare standards in China, it should present solidly based information, in bibliographic sources and/or in the authors' own experience.

However, for me, it falls short of expectations, with many statements of subjective content and/or without proper bibliographic references or source, which can compromise its credibility with stakeholders in the poultry industry in China. Changes by authors after the first review did not significantly improve the scientific quality of the review.

In the PDF with the manuscript that I have attached, I suggest some mainly "cosmetic" changes and point to some fundamental problems, but even so, these are only a small fraction of what should be done to substantially improve this review.

Regarding the free range systems (4.1, 4.2 and 4.3), I suggest summarizing the particularities in a table, to compare systems more easily.

Reviewer 2 Report

Dear Authors,

Many thanks for submitting this revised version of the manuscript for review. You have taken into account the feedback provided on the initial review of the paper. The hollistic consideration of the farming situation for the chickens is now much clearer.. The developments to the manuscript have resulted in a more robust paper overall. In light of the revisions, the paper is now in a much better position for consideration.

Round 3

Reviewer 1 Report

The manuscript improved after the 2nd revision. However, there are still some aspects that must be addressed. In this regard, I have included several suggestions in the attached PDF.
